# Segmentation of Lipid Droplets in Histological Images

**Daniel Budelmann**[1]                    DANIEL.BUDELMANN@MEVIS.FRAUNHOFER.DE

**Cao Qing**[2]                                    CAOQINGCF@GMAIL.COM

**Hendrik Laue**[3]                              HENDRIK.LAUE@MEVIS.FRAUNHOFER.DE

**Mohamed Albadry**[4,5]                  MOHAMED.ALBADRY@MED.UNI-JENA.DE

**Uta Dahmen**[4]                              UTA.DAHMEN@MED.UNI-JENA.DE

**Lars Ole Schwen**[3]                        OLE.SCHWEN@MEVIS.FRAUNHOFER.DE

[1] *Fraunhofer Institute for Digital Medicine MEVIS, Lübeck, Germany*

[2] *Machine Learning for Computer Vision, TU Dresden, Dresden, Germany*

[3] *Fraunhofer Institute for Digital Medicine MEVIS, Bremen, Germany*

[4] *Experimental Transplantation Surgery, Department of General, Visceral and Vascular Surgery, University Hospital Jena, Jena, Germany*

[5] *Department of Veterinary Pathology, Menoufia University, Egypt*

## Abstract

Steatosis is a common liver disease characterized by the accumulation of lipid droplets in cells. Precise and reliable fat droplet identification is essential for automatic steatosis quantification in histological images. We trained a nnU-Net to automatically segment lipid vacuoles in whole-slide images using semi-automatically generated reference annotations. We evaluated the performance of the trained model on two out-of-distribution datasets. The trained model's average F1 scores (0.801 and 0.804) suggest a high potential of the nnU-Net framework for the automatic segmentation of lipid vacuoles.

**Keywords:** Hepatic steatosis, histology, whole-slide images, nnU-Net, segmentation

## 1. Introduction

Steatosis, the fat accumulation in liver cells, is the predominant symptom of alcoholic and non-alcoholic fatty liver disease.

Quantification of steatosis is an important factor in the decision to proceed with liver transplantation, and visual inspection of tissue stained with hematoxylin and eosin (H&E) is a common method (Roy et al., 2020).

Semi- and fully-automatic image segmentation approaches have been developed for computer-aided quantification of steatosis (Homeyer et al., 2015; Roy et al., 2020). Training machine learning methods for this purpose requires annotated image data, which is tedious to create manually.

We explored an approach using semi-automatically generated and thus imperfect data to train an off-the-shelf medical image segmentation method, namely the self-configuring nnU-Net (Isensee et al., 2021).

## 2. Datasets and Methods

**Image Datasets**  Besides one dataset (A) used for training and in-distribution evaluation, we evaluated the performance on two out-of-distribution datasets to assess intra- (B) and inter-species (C) generalizability. *Dataset A*[1] consists of 19 whole-slide images (WSI) of H&E-stained tissue from male C57BL/6J mice with diet-induced steatosis of different severity.

*Dataset B*[2] consists of 36 WSI of H&E-stained slides of one male C57BL/6N mouse with diet-induced steatosis. The images of datasets A and B have a resolution of 908 nm/pixel.

*Dataset C*[3] contains H&E-stained human liver tissue scanned at $20\times$ objective magnification and contains pixel-level annotations (exact resolution not specified).

**Reference Data Preparation**  Dataset A was annotated by a non-expert using the semi-automatic approach by Homeyer et al. (2015); images and segmentation masks were subsequently tiled in $256\times256$ images. For training and validation, 16 WSI and corresponding annotations of dataset A were used, three (one each for every steatosis extent) were kept back as a test set. Dataset B was annotated in the same way as dataset A, a randomly selected subset of segmentation masks was subsequently corrected manually by visual inspection using GIMP-2.10. Reference segmentations for datasets A and B are available from https://doi.org/10.5281/zenodo.7802210.

To obtain compatible image size and resolution for dataset C, we mirrored and concatenated these patches in both dimensions and downsampled them subsequently. This introduced artifacts (mirrored partial cells and lipid vacuoles) near the stitching boundary. To avoid these artifacts in the evaluation, a border of 25 pixels (approximately 25 μm, the average size of the structures of interest) extending from the mirroring axis was omitted.

**Training and Evaluation**  We trained the 2D nnU-Net (Isensee et al., 2021) on the generated reference dataset A using all three color channels, for 1000 epochs, and with five different folds of training/validation split.

We quantified segmentation accuracy of the trained nnU-Net by a pixel-wise F1 score over all tiles containing tissue. This prevents irrelevant (background-only) regions from artificially simplifying the task. Code is available from https://doi.org/10.5281/zenodo.7802210.

## 3. Results and Discussion

Trained on the data generated using a semi-automatic segmentation method with minimal effort, the nnU-Net generalized well to different datasets from the same and from a different species (F1 scores 0.732 and 0.804, respectively, see Table 1). A higher F1 score for the corrected subset of B (0.744 vs. 0.801) indicates that the nnU-Net reflects human vision better than the semi-automatic approach. The nnU-Net identifies smaller droplets as fat, whereas the expert annotation in dataset C does not, see Figure 1.

---

1. available from https://doi.org/10.15490/FAIRDOMHUB.1.STUDY.1070.1, (Albadry et al., 2022)

2. available from https://doi.org/10.5281/zenodo.4738561, (Budelmann et al., 2022)

3. available from https://figshare.com/s/d75b129d969b4f463168, (Roy et al., 2020)

Table 1: Mean F1 scores for the segmentation result of the trained nnU-Net models

| Dataset | # Patches | F1 Score |
|---|---|---|
| Test A | 4512 | 0.867 |
| B | 65176 | 0.732 |
| subset B | 232 | 0.744 |
| subset B, corrected | 237 | 0.801 |
| C | 736 | 0.804 |

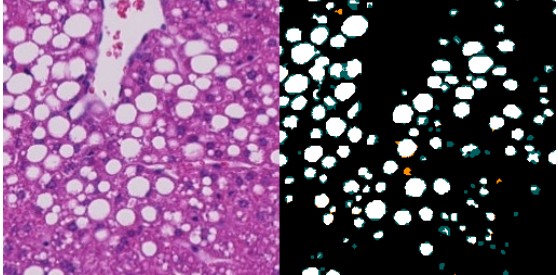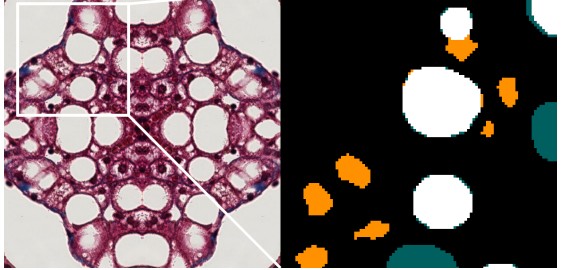

Figure 1: Left: Example tile of corrected subset B with a visualization of true negative (black), true positive (white), false negative (teal) and false positive (orange) pixels; right: Example tile of dataset C with evaluation area

## Acknowledgments

This work was funded by Deutsche Forschungsgemeinschaft (DFG) via project 410848700 (SteaPKMod).

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
