# OpenReview forum: "Segmentation of Lipid Droplets in Histological Images"
_MIDL.io/2023/Short_Paper_Track — MIDL 2023 Short paper track Poster_

### Official Review · Reviewer_Kn7t · 2023-04-10
**nnU-Net for pathological image segmentation**

**Rating:** 6
**Confidence:** 5

**Review:**

This paper trained a nnU-Net to automatically segment lipid vacuoles in whole-slide images.
The advantages of the paper include:
It provides source code, qualitative results, and quantitative results in the short paper.
This is an interesting clinical application.
The limitation of the paper includes:
No baseline methods are included.
The methodological innovation is limited as the paper applies an existing segmentation model.

---

### Official Review · Reviewer_jTFT · 2023-04-25
**segmentation of lipid vacuoles in whole-slide histology images using semi-automatically generated imperfect reference annotations**

**Rating:** 5
**Confidence:** 4

**Review:**


The short paper train a nnU-Net for segmentation of lipid
vacuoles in whole-slide histology images using semi-automatically generated imperfect reference annotations. They evaluated the performance on two out-of-distribution datasets. The trained model had F1 scores of 0.801 and 0.804 using the nnU-Net framework for the automatic segmentation of lipid vacuoles. The results are promising but the methods are quite standard.